# Relationships between Abyssal Redox Conditions and Rock Magnetic Properties of Surficial Sediments in the Western Pacific

**Yanping Chen** [1,†]**, Dong Xu** [2,†]**, Huafeng Qin** [3,4]**, Geng Liu** [3,4]**, Yibing Li** [5]**, Weiwei Chen** [5] **and Liang Yi** [5,*]

1   Zhejiang Academy of Marine Sciences, Second Institute of Oceanography, Ministry of Natural Resources, Hangzhou 310012, China; chenai0812@163.com
2   MNR Key Laboratory of Submarine Geosciences, Second Institute of Oceanography, Ministry of Natural Resources, Hangzhou 310012, China; xudongsio@126.com
3   State Key Laboratory of Lithospheric Evolution, Institute of Geology and Geophysics, Chinese Academy of Sciences, Beijing 100029, China; huafeng_qin@mail.iggcas.ac.cn (H.Q.); gengliu1995@outlook.com (G.L.)
4   University of Chinese Academy of Sciences, Beijing 100049, China
5   State Key Laboratory of Marine Geology, Tongji University, Shanghai 200092, China; yibing_l@163.com (Y.L.); sww@tongji.edu.cn (W.C.)
*   Correspondence: yiliang@tongji.edu.cn
†   These authors contributed equally to this work.

**Abstract:** Reconstructing changes in deep/bottom-water redox conditions are critical for understanding the role of the deep ocean in global carbon and metals cycling; nevertheless, the quantitative relationships between redox proxies and abyssal dissolved oxygen are poorly investigated. In this work, we studied the rock magnetic properties of surficial sediments in the western Pacific to investigate their relationship with regional redox conditions. Our results reveal a consistent sedimentary magnetic mineral assemblage in the western Pacific, dominated by pseudo-single-domain magnetite ($Fe_3O_4$), while the ratio of detrital and biogenic magnetite particles in different sites varies substantially. Detailed analyses identified two major magnetic-coercivity components, with modal coercivity values of $13.1 \pm 1.6$ mT and $54.7 \pm 5.3$ mT, respectively. All the magnetic parameters we measured, including both concentration-dependent and grainsize-dependent parameters, and the magnetic coercivities, are generally correlated to sedimentary redox conditions; however, the coercivities obtained by mathematical unmixing exhibit a stronger linkage, explaining about a quarter of variance of redox changes. Our findings confirm the potential of magnetic properties for tracing abyssal redox changes in the western Pacific, while the observed magnetic-redox relationships are complex and need further investigation.

**Keywords:** deep-sea sediments; magnetic properties; magnetic coercivity; redox changes; western Pacific

## 1. Introduction

Redox conditions are a critical environmental factor in paleoceanography, e.g., [1–4], representing a key interaction point between various climatic system components, such as oceanic water masses, marine productivity, and sediment fluxes. To reconstruct the past changes in redox conditions, the proxies are mainly based on geochemical properties of foraminiferal shells or deep-sea sediments. For example, for Cenozoic sediments, the geochemical properties of benthic foraminifera are often employed. One of the most-used proxies is the stable carbon isotope composition ($\delta^{13}C$) and its inter-basinal gradient ($\Delta\delta^{13}C$), e.g., [5–7]. Carbon isotopes in ocean water are associated with the balance between dissolved inorganic and organic carbon, linked to deep-sea ventilation and thus redox conditions [8], while other studies have indicated that seawater temperature and carbon species have more influence on benthic $\delta^{13}C$ [9,10]. Trace elements in foraminifera and/or

sediments, including Mn, Zn, Ni, and V, e.g., [1,11,12], generally migrate from oxidizing to reducing environments; however, migration in the opposite direction may also occur, in the case of authigenic uranium, e.g., [3,13] and some rare earth elements (Ce, Y, and Ho), e.g., [14,15]. Thus, these trends reflect changes in redox conditions. Sedimentary and/or foraminiferal neodymium isotopes ($^{143}$Nd/$^{144}$Nd) are also used to trace redox changes, e.g., [16,17], because different water masses have distinct neodymium isotope patterns [18], which reflect changes in deep-sea ventilation and redox conditions. However, most present studies using the geochemical properties of foraminiferal shells or abyssal sediments are qualitative or semiquantitative, and there is a lack of quantitative tests, partially due to the high time- and expense-cost of geochemical measurements.

As a proxy that can be rapidly and easily measured, the magnetic properties of deep-sea sediments were recently proposed to be associated with deep-water redox conditions, e.g., [19–26]. Magnetic minerals are redox sensitive, due to conversions between $Fe^{2+}$ and $Fe^{3+}$ in crystal lattices, as in the cases of magnetite, maghemite, hematite, and iron sulfides (greigite or pyrrhotite). For example, experimental investigations of magnetic particles with different oxidation states showed that pseudo-single-domain (PSD) magnetite was significantly negatively correlated to partial oxidation in a shell-only model [27]. Thus, either in paleoenvironmental or experimental contexts, the potential of magnetic properties for reconstructing deep-sea redox conditions has been proposed. Nevertheless, there are few quantitative evaluations of magnetic properties–redox relationships using sediment samples covering a large region.

Given the absence of carbonate fossils beneath the carbonate compensation depth, which comprises a large area of the Pacific, it is important to further develop sedimentary proxies for deep-sea oxidation conditions. To this end, we compared the spatial distribution of the magnetic properties of bottom sediments from the western Pacific with abyssal (sedimentary) oxidation conditions in this study.

## 2. Materials and Methods

### 2.1. Study Sites

Sediment samples from 66 sites were collected in the western Pacific (Figure 1) in October 2017, using a box corer. The samples were obtained from the depth range of 2630–6060 m, and the sediments were predominantly light-brown to brown muds. The redox potential (Eh value) of the sediments was immediately measured on board, using a portable pH meter (INESA Co., PHBJ-260). Since the measurements were conducted in an on-board lab, the obtained Eh values may not directly represent the real ones of the deep-sea environment, but instead a proxy of sedimentary redox conditions.

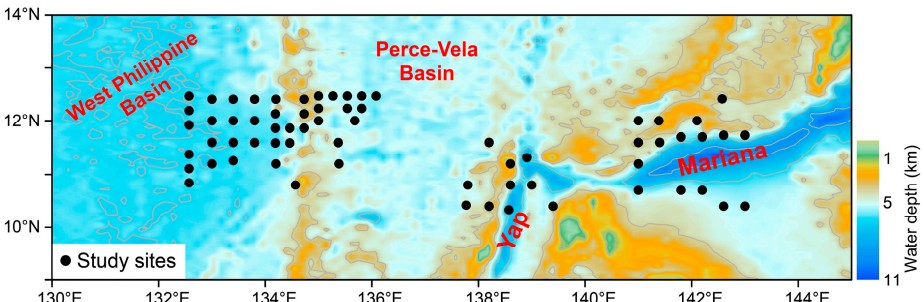

**Figure 1.** Schematic map showing the study area and study sites. Mariana, Mariana Trench; Yap, Yap Trench. The site information is listed in Table 1.

**Table 1.** The information of the 66 study sites.

| Sites | Longitude (°E) | Latitude (°N) | Water Depth (m) | Sites | Longitude (°E) | Latitude (°N) | Water Depth (m) |
|---|---|---|---|---|---|---|---|
| A14 | 137.77 | 10.42 | 4690 | E04 | 133.80 | 12.00 | 5720 |
| A15 | 138.20 | 10.40 | 4200 | E07 | 135.00 | 12.00 | 4620 |
| A16 | 138.57 | 10.33 | 5990 | E09 | 135.68 | 12.00 | 4750 |
| A18 | 139.40 | 10.40 | 3890 | E22 | 141.00 | 12.00 | 4070 |
| J01A | 142.28 | 10.35 | 4430 | E23 | 141.38 | 12.00 | 2630 |
| A27 | 142.60 | 10.40 | 4920 | E25 | 142.10 | 12.00 | 4120 |
| B06 | 134.57 | 10.80 | 4025 | F02 | 133.00 | 12.40 | 5540 |
| B14 | 137.80 | 10.80 | 4230 | F03 | 133.40 | 12.40 | 5850 |
| B16 | 138.60 | 10.80 | 5340 | F04 | 133.80 | 12.40 | 5220 |
| B17 | 139.00 | 10.80 | 3540 | F05 | 134.20 | 12.40 | 5180 |
| B22 | 141.00 | 10.70 | 5605 | F26 | 142.57 | 12.41 | 3560 |
| B24 | 141.80 | 10.70 | 4800 | G01 | 132.57 | 12.46 | 5830 |
| B25 | 142.20 | 10.70 | 5090 | G09 | 134.73 | 12.40 | 3390 |
| C02 | 133.00 | 11.20 | 5720 | G10 | 135.00 | 12.46 | 4040 |
| C03 | 133.40 | 11.26 | 5710 | G11 | 135.27 | 12.46 | 4610 |
| C05 | 134.20 | 11.20 | 5330 | G12 | 135.54 | 12.46 | 4880 |
| C08 | 135.38 | 11.20 | 4200 | G13 | 135.81 | 12.46 | 5041 |
| C16 | 138.60 | 11.20 | 3120 | G14 | 136.08 | 12.46 | 5063 |
| C17 | 138.91 | 11.32 | 5670 | G15 | 132.57 | 12.19 | 6060 |
| C22 | 141.00 | 11.20 | 4780 | G21 | 134.19 | 12.13 | 5020 |
| D02 | 133.00 | 11.60 | 5950 | G23 | 134.73 | 12.13 | 3430 |
| D03 | 133.40 | 11.60 | 5630 | G24 | 135.00 | 12.23 | 3680 |
| D04 | 133.80 | 11.60 | 5230 | G26 | 135.54 | 12.23 | 4770 |
| D08 | 135.37 | 11.60 | 4260 | G27 | 135.81 | 12.23 | 5100 |
| D15 | 138.20 | 11.60 | 4180 | G28 | 132.57 | 11.92 | 5970 |
| D22 | 141.00 | 11.60 | 3310 | G34 | 134.19 | 11.86 | 5510 |
| D23 | 141.40 | 11.60 | 4100 | G35 | 134.46 | 11.86 | 3330 |
| D24 | 141.80 | 11.70 | 5370 | G36 | 134.73 | 11.86 | 4940 |
| D25 | 142.20 | 11.70 | 5590 | G46 | 134.19 | 11.59 | 5580 |
| D26 | 142.60 | 11.73 | 5020 | G47 | 134.46 | 11.59 | 4750 |
| D27 | 143.00 | 11.73 | 3960 | G51 | 132.57 | 11.38 | 5770 |
| E02 | 133.00 | 12.00 | 5880 | G57 | 132.57 | 11.11 | 5820 |
| E03 | 133.40 | 12.00 | 5640 | G60 | 132.57 | 10.84 | 5790 |

## 2.2. Magnetic Measurements

For the measurement of magnetic properties, the samples collected from the 66 sites were dried in vacuum and then packed in 8 cm$^3$ cubic plastic boxes. Magnetic susceptibility was measured using an AGICO MFK1-FA Multi-Frequency Kappabridge magnetic susceptibility meter, at frequencies of 976 Hz and 15,616 Hz ($\chi$LF and $\chi$HF, respectively). Anhysteretic remanent magnetization (ARM) was imparted using a peak alternating field (AF) of 100 mT and a direct biasing field of 0.05 mT using a 2G Enterprises SQUID magnetometer with inline AF coils. The susceptibility of ARM ($\chi$ARM) was calculated as ARM/0.05 mT. Isothermal remanent magnetization (IRM) was imparted with a 2G Enterprises model 660 pulse magnetizer, with successive pulsed fields of 1 T (saturation IRM, SIRM), −0.1 T (IRM$_{-0.1T}$), and −0.3 T (IRM$_{-0.3T}$). The temperature-dependent magnetic susceptibilities ($\chi$–T curves) were measured by continuous exposure of samples through temperature cycles from room temperature to 700 °C and back to room temperature in an argon atmosphere, using a Kappabridge KLY-3 with a CS-3 high-temperature furnace (Agico Ltd., Brno, Czech Republic). These measurements were conducted in the State Key Laboratory of Marine Geology, Tongji University.

The percentage frequency-dependent magnetic susceptibility ($\chi$fd), and Sratio were calculated as $\chi_{fd} = \frac{\chi_{LF} - \chi_{HF}}{\chi_{LF}} \times 100\%$ and $S_{ratio} = -\frac{\text{IRM}_{-0.3T}}{\text{SIRM}} \times 100\%$, respectively. To assess the extent of early diagenetic alteration of the magnetic minerals, hysteresis loops and IRM acquisition curves were measured on all samples, using a MicroMag 3900 Vibrating Sample

Magnetometer (VSM, Princeton Measurements Inc., Westerville, OH, USA), with a peak field of 0.3 T. Saturation magnetization (Ms), saturation remanence (Mrs), coercive force (Bc), and the coercivity of remanence (Bcr) were determined from the hysteresis loops, after correction using the data between 0.25 and 0.30 T.

The mathematical unmixing of hysteresis loops and IRM acquisition curves can provide detailed information about different coercivity spectra [28]. A series of target functions have been proposed for unmixing [29]. We used the normal function to identify the potential end members of magnetic minerals in the sediments [29,30].

The polymodal distribution for unmixing in this study is expressed as follows:

$$F = p_1 f_1 + \cdots + p_i f_i; \sum_1^n p_i = 1 \tag{1}$$

where *fi* represents the function for component *i* where *i* = 1 to *n* components, and $p_i$ is the percentage contribution of components. The normal function has the following form:

$$= p_1 \frac{1}{\sqrt{2\pi}\alpha_1} e^{-\left[\frac{(x-\beta_1)^2}{2\alpha_1^2}\right]} + (1-p_1) \frac{1}{\sqrt{2\pi}\alpha_2} e^{-\left[\frac{(x-\beta_2)^2}{2\alpha_2^2}\right]} \tag{2}$$

Here, $x$ is the independent variable and represents the magnetic field (IRMs on a logarithmic scale), and the dependent variables are the first and second derivatives of the IRM acquisition and hysteresis loop data, respectively, which were first standardized to the interval of [0, 1]. The coefficient $p$ represents the relative ratio between two components (namely cont1 and cont2), $\alpha$ determines the distribution shape, and $\beta$ controls the position of the central tendency of the curve, herein, the magnetic coercivity (namely Coe1 and Coe2).

First-order reversal curves (FORCs) were determined for five representative samples, setting a peak field of 1.0 T and an interval of 3.2 mT. To further characterize the magnetic minerals, a magnetic extract was obtained from a representative sample (from Site J01A) and used for transmission electron microscope (TEM) analysis, with a JEOL JEM2100F LaB6. These measurements were conducted in the Paleomagnetism and Geochronology Lab (PGL), Institute of Geology and Geophysics, Chinese Academy of Sciences.

## 3. Results

### 3.1. Relationships between Magnetic Parameters

As shown in Table 2, the mass-specific χLF of the samples varies from $1.4 \times 10^{-7}$ m$^3$/kg to $7.8 \times 10^{-6}$ m$^3$/kg, and χfd is generally high (>5%) and variable (mean and standard deviation of $8.2 \pm 7.2\%$). χARM is sensitive to single domain (SD) grains [31,32], and SIRM reflects the content of the remanence-bearing fraction, and it excludes the influence of superparamagnetic (SP) grains [33]. The ranges for χARM and SIRM are $3.8 \times 10^{-7}$–$6.7 \times 10^{-5}$ m$^3$/kg and 0.28–$9.6 \times 10^{-3}$ Am$^2$/kg, respectively.

**Table 2.** Statistic properties of magnetic parameters of the 66 study sites.

|  | Units | Min | Max | Mean | Standard Deviation | Skewness | Kurtosis |
|---|---|---|---|---|---|---|---|
| χLF | m$^3$/kg | $1.4 \times 10^{-7}$ | $7.8 \times 10^{-6}$ | $1.9 \times 10^{-6}$ | $1.4 \times 10^{-6}$ | 1.5 | 3.6 |
| χHF | m$^3$/kg | $1.2 \times 10^{-7}$ | $7.5 \times 10^{-6}$ | $1.8 \times 10^{-6}$ | $1.4 \times 10^{-6}$ | 1.5 | 3.4 |
| χfd | % | 1.1 | 33.1 | 8.2 | 7.2 | 2.1 | 4.0 |
| χARM | m$^3$/kg | $3.8 \times 10^{-7}$ | $6.7 \times 10^{-5}$ | $1.2 \times 10^{-5}$ | $1.1 \times 10^{-5}$ | 2.8 | 10.8 |
| SIRM | Am$^2$/kg | $2.8 \times 10^{-4}$ | $9.6 \times 10^{-3}$ | $2.8 \times 10^{-3}$ | $1.9 \times 10^{-3}$ | 1.0 | 1.2 |
| IRM$_{-0.1T}$ | Am$^2$/kg | $1.6 \times 10^{-4}$ | $4.7 \times 10^{-3}$ | $1.8 \times 10^{-3}$ | $1.3 \times 10^{-3}$ | 0.7 | −0.6 |
| IRM$_{-0.3T}$ | Am$^2$/kg | $2.7 \times 10^{-4}$ | $9.4 \times 10^{-3}$ | $2.7 \times 10^{-3}$ | $1.9 \times 10^{-3}$ | 1.0 | 1.1 |

The concentration-dependent magnetic parameters (χ, χARM, and SIRM) show similar variations. Considering the general relationships between the magnetic parameters

(Figure 2, Table 3), we suppose that there are no significant differences in magnetic properties (hence in magnetic mineral sources) between the study sites. However, the large variations of the concentration-dependent magnetic parameters indicate large differences in the magnetic mineral content of these sediments, suggesting that the dataset may be broadly representative of the bottom sediments in the western Pacific.

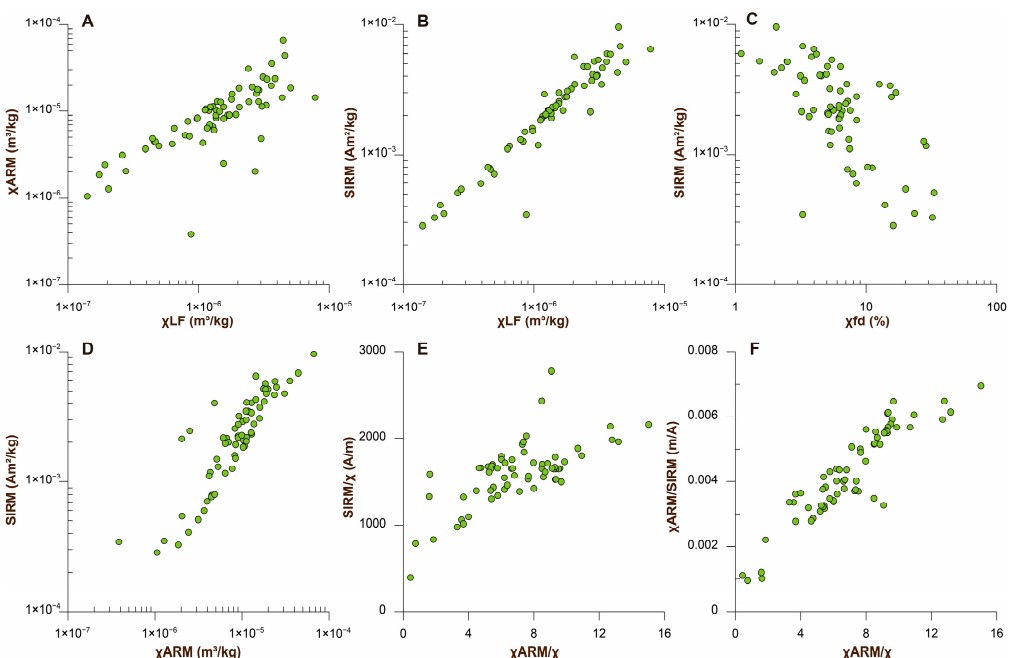

**Figure 2.** Scatterplot of the relationships between selected magnetic parameters (**A**–**F**). It is noted that there were in total 66 samples analyzed.

**Table 3.** Correlation coefficients between chemical elements of the 66 study sites.

|  | χLF | χfd | χARM | SIRM | χARM/χ | SIRM/χ |
|---|---|---|---|---|---|---|
| χLF | 1 |  |  |  |  |  |
| χfd | −0.48 | 1 |  |  |  |  |
| χARM | 0.64 | −0.38 | 1 |  |  |  |
| SIRM | 0.88 | −0.51 | 0.87 | 1 |  |  |
| χARM/χ | −0.32 | 0.32 | 0.34 | −0.04 [1] | 1 |  |
| χARM/SIRM | −0.22 [1] | 0.24 [1] | 0.36 | −0.04 [1] | 0.91 | 1 |
| SIRM/χ | −0.38 | 0.25 | 0.14 [1] | 0.57 | 0.69 | 0.39 |

[1] All coefficients are significant at $p < 0.05$ level, except for these five values. The correlation coefficients are all based on a linear relationship.

### 3.2. Spatial Distribution of Magnetic Parameters

The consistency of magnetic properties between the sites in the western Pacific offers the opportunity to visualize the spatial pattern of individual magnetic parameters and to compare them with the Eh values.

As can be seen in Figures 3 and 4, the Eh values are relatively high in the West Philippine Basin and the Parece Vela Basin and low at the sites from the Yap and Mariana trenches. The spatial pattern of the Eh values can be attributed to the generally deeper water depths of the study sites in the two basins, where the dissolved oxygen levels are maintained by the Antarctic bottom water (AABW) [34,35] and/or the unique submarine topography [36,37].

For the concentration-dependent magnetic parameters (χ, χARM, and SIRM), high values occur in the vicinity of the two trenches, while in the two basins, relatively low values are found (Figure 3). The cause of this spatial pattern can partially be attributed to

water depth, since shallower sites in the western Pacific may be more influenced by marine productivity in the upper ocean layer, e.g., [37–39].

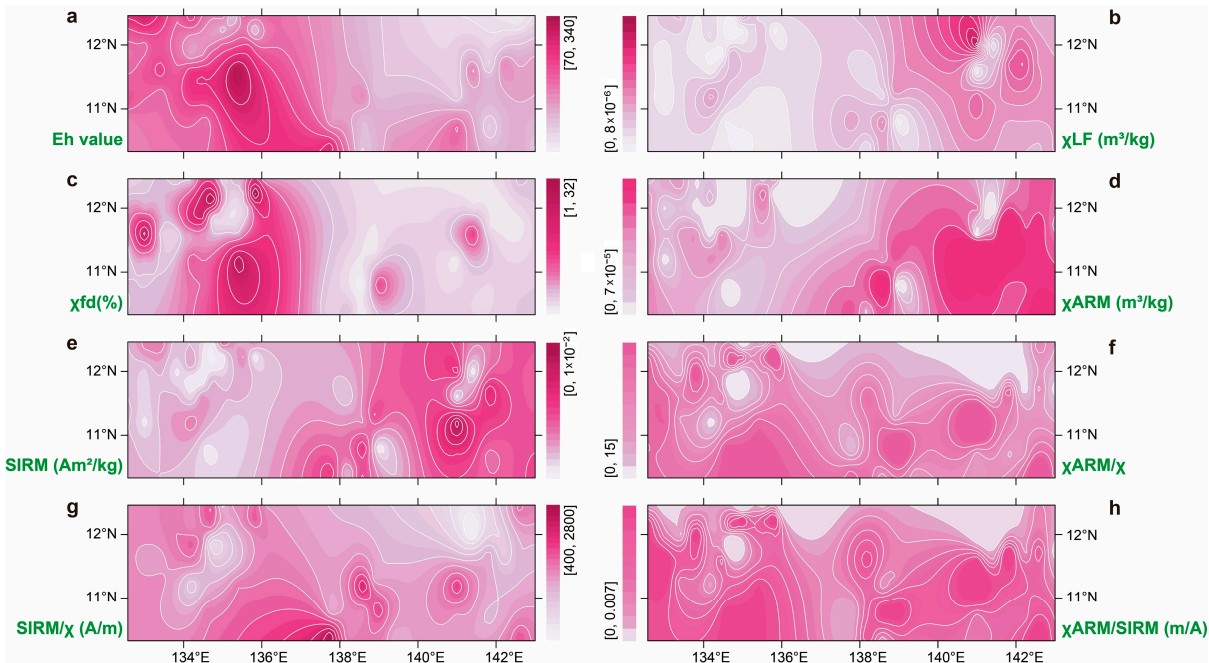

**Figure 3.** Spatial distribution of Eh values and the selected magnetic parameters (**a**–**h**).

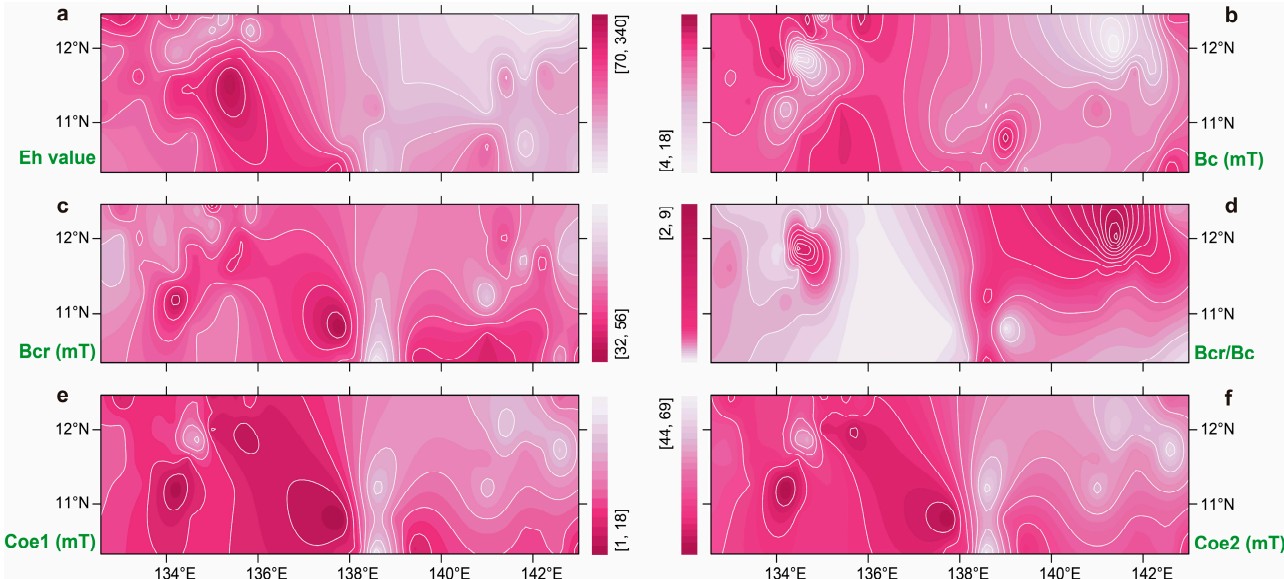

**Figure 4.** Spatial distribution of Eh value, Bcr/Bc, and magnetic coercivities (**a**–**f**). Note that the scales in (**c**,**e**,**f**) are reversed.

For the grainsize-dependent rock magnetic parameters ($\chi$fd, $\chi$ARM/$\chi$, SIRM/$\chi$, and $\chi$ARM/SIRM), no consistent spatial pattern is evident (Figure 3), potentially indicating a complex response of the magnetic grain size to specific regional settings. However, a similarity between magnetic coercivities and Eh is observed (Figure 4), that is, coercivity values are relatively high in the Yap and Mariana trenches, but low in the two basins. This spatial pattern of magnetic coercivity in abyssal sediments is consistent with the results of experimental studies, e.g., [27], which generally indicate that the magnetic properties of PSD magnetite are sensitive to early stage oxidation.

### 3.3. Sedimentary Magnetic Mineralogy

The magnetic hysteresis loops are closed below 200 mT (Figure 5a), and the IRM acquisition curves are saturated in relatively low magnetic fields (<200 mT) (Figure 5d), with Sratio values of 0.945–0.998, indicating that these sediments are dominated by low-coercivity magnetic minerals.

The Bc and Bcr values are 13.5 ± 2.7 mT and 43.7 ± 4.2 mT (Table 4), respectively, and on the Day plot [40], they are distributed within the PSD range (Figure 5c). A two-humped distribution of the normal function was applied to mathematically unmix the hysteresis loops [29,30], and their coercivities (Coe1 and Coe2) are 13.1 ± 1.6 mT and 54.7 ± 5.3 mT, respectively (Figure 5b; Table 4). The two-humped distribution in the log-normal function was also applied to unmix the IRM acquisition curves [41]. There is no significant difference in the Coe2 values between the hysteresis-loop-based and IRM-acquisition-based results (r = 0.58, *p* < 0.01) (Figure 5e), confirming the reliability of the mathematical unmixing. In addition, in χ–T heating curves, a gradual decrease in magnetic susceptibility till ~580 °C was observed (Figure 5g–i).

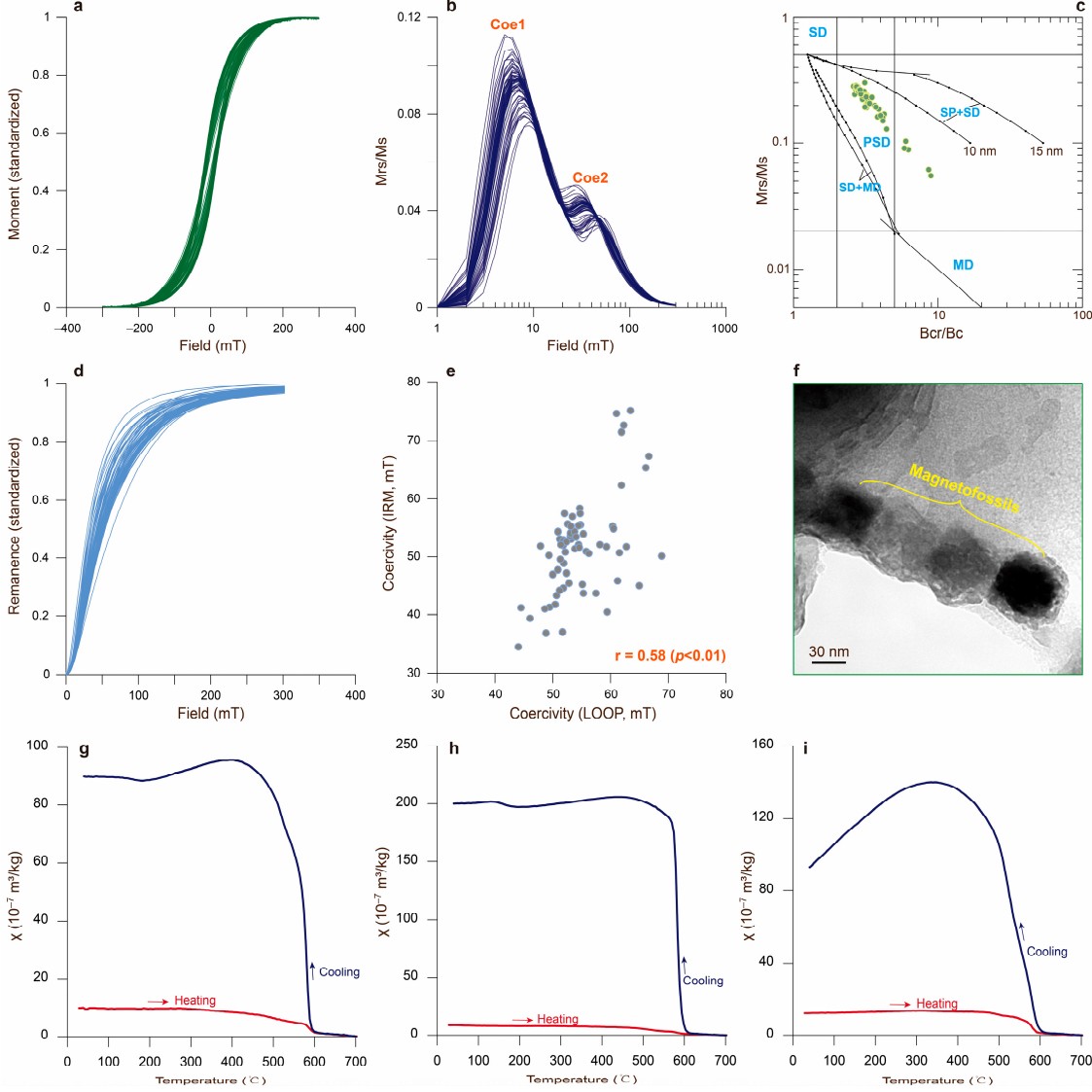

**Figure 5.** Rock magnetic properties of the bottom sediment samples from the western Pacific. (**a**) Hysteresis loops (calibrated). (**b**) Mathematical unmixing of the second-order derivative of the

hysteresis loops showing two coercivity components (Coe1 and Coe2). (**c**) Day plot [42,43]. SD, single domain; MD, multidomain; PSD, pseudo-single-domain; SP, superparamagnetic. (**d**) IRM acquisition curves. (**e**) Comparison between the hysteresis-loop-based and IRM-acquisition-based results. It is noted that there were in total 66 samples analyzed in (**a**,**d**). (**f**) TEM image of probable magnetofossils (Site J01A, Mariana Trench). (**g–i**) χ–T curves of sites E03 (West Philippine Basin), C16 (Yap Trench), and J01A (Mariana Trench), respectively.

**Table 4.** Statistic properties of loop parameters of the 66 study sites.

|  | Units | Min | Max | Mean | Standard Deviation | Skewness | Kurtosis |
|---|---|---|---|---|---|---|---|
| Bc | mT | 4.5 | 17.1 | 13.5 | 2.7 | −1.4 | 1.9 |
| Bcr | mT | 32.0 | 55.5 | 43.6 | 4.2 | −0.6 | 1.2 |
| Mrs/Ms | - | 0.055 | 0.29 | 0.22 | 0.05 | −1.3 | 1.3 |
| Cont1 | % | 49.2 | 57.7 | 52.8 | 2.1 | 0.6 | −0.2 |
| Coe1 | mT | 9.9 | 17.7 | 13.1 | 1.6 | 0.7 | 0.3 |
| Cont2 | % | 42.3 | 50.8 | 47.2 | 2.1 | −0.6 | −0.2 |
| Coe2 | mT | 44.0 | 68.9 | 54.7 | 5.3 | 0.6 | 0.1 |

Additionally, there is a transition from a large vertical spread to a more limited spread in the contours in the FORC diagrams (Figure 6), with most of the coercivity distribution within the range of 10–60 mT, and a peak at ~20 mT. The limited vertical spread and wide horizonal spread of the contours (Figure 6d,e) indicate minimal magnetostatic interactions [44,45], which is in agreement with comparable rock magnetic results from the Mariana Trench [46] and the West Philippine Basin [36].

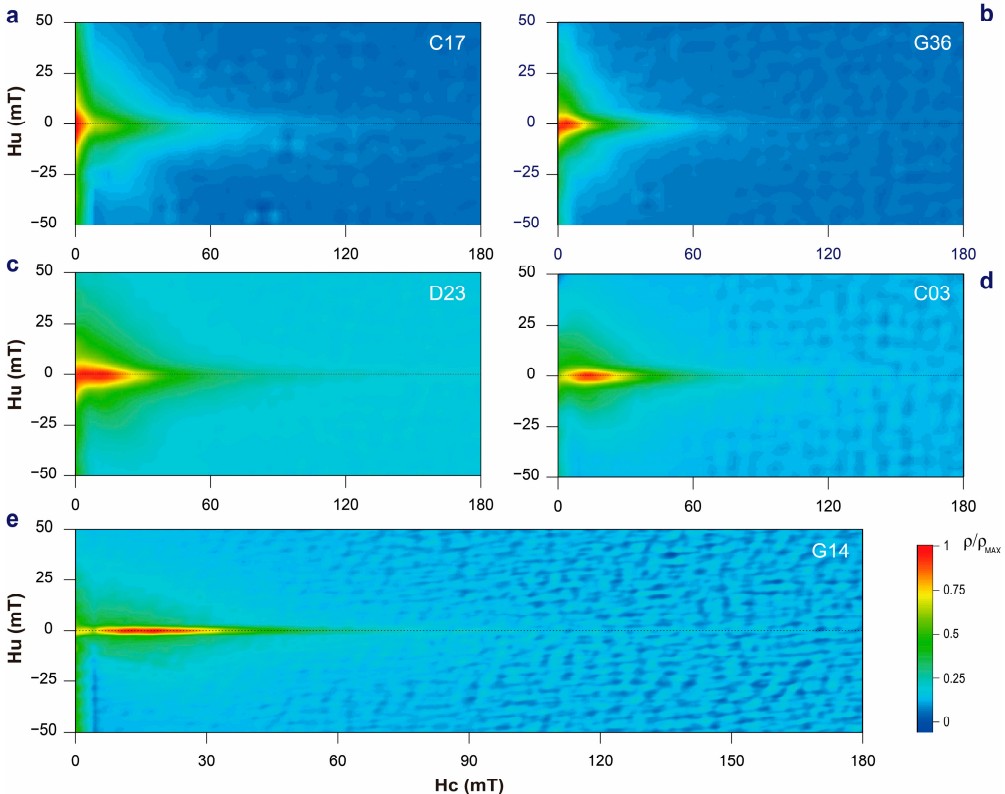

**Figure 6.** FORC diagram for five representative samples. All diagrams were produced using the FORCme software with a smoothing factor of 3. Sample G14 (**e**) was subjected to high-resolution FORC analysis (360 lines), while for the other samples (**a–d**), a standard analysis was conducted (90 lines).

The observed magnetic characteristics of the study sites are similar to those of cores NGC36, NGC65, and NGC88 from the western Pacific [38], core MABC-11 from the Magellan Seamount [37], and core XTGC1311 from the middle Pacific [47], for which the presence of the magnetofossils from magnetic bacteria were claimed. The TEM investigation of Site J01A (Figure 5f) shows a chain-like shape of magnetic minerals, confirming that magnetofossils are widespread in the abyssal sediments of the western Pacific. Thus, we propose that the transition in the FORC distribution evident in Figure 6a–e likely represents an increase in the content of magnetofossils in the sediments.

## 4. Discussion

Integrating the foregoing observations with previous studies in this region [11,38,39,46,48], it is inferred that magnetite ($Fe_3O_4$) is the dominant magnetic mineral in studied sediments, and that our samples are representative of the magnetic properties of the bottom sediments of the western Pacific. Thus, they offer the opportunity to test the relationship between deep-water redox conditions and sediment magnetic properties (Figure 7).

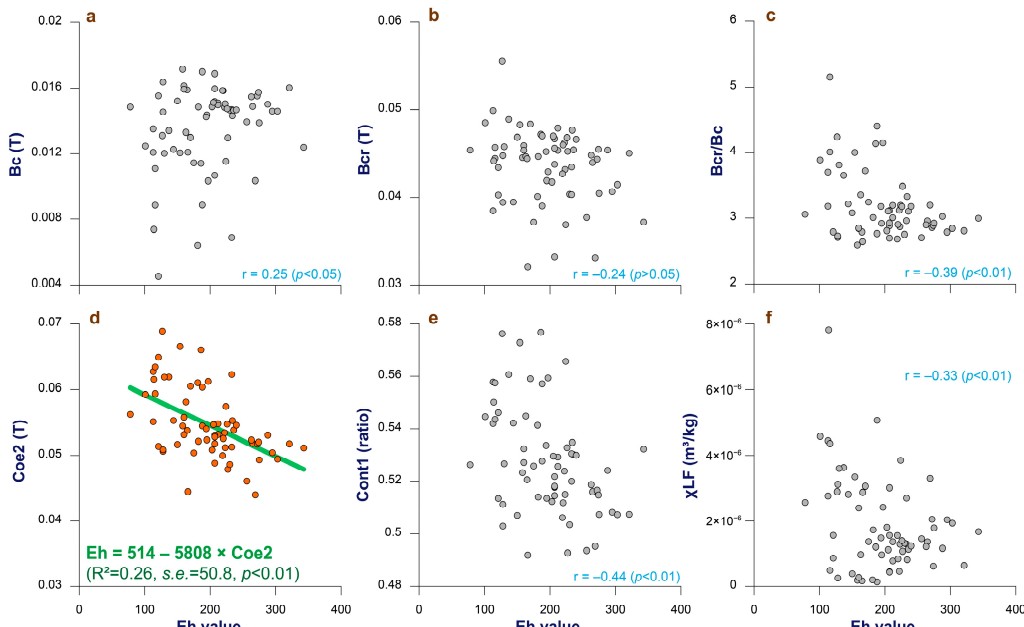

**Figure 7.** Results of regression analysis between selected rock magnetic parameters and deep-sea redox conditions (Eh value). Correlation coefficients are included in all plots (**a–f**), and the best-fit regression line is shown in (**d**).

The relationship is weak between Eh and Bc and Bcr, which is only significant at the 0.05 level (Figure 7a,b). This weak correlation suggests that redox conditions were not the only factor significantly influencing the bulk magnetic coercivity, and that other factors may include different proportions of magnetite grains with different coercivities. This inference is supported by the higher correlation between Eh and Coe2 and Cont1 (Figure 7d,e). In addition, different grain sizes of magnetic minerals may have different behaviors responding to oxidation [27], which could explain a weak correlation between Eh and Bcr/Bc (Figure 7c), since the Bcr/Bc ratio can be a proxy of magnetic grain size [49], like $\chi$fd, $\chi$ARM/$\chi$, SIRM/$\chi$, and $\chi$ARM/SIRM.

We then conducted stepwise linear regression analysis of these magnetic parameters, and only the parameter Coe2 was retained in the analysis (Figure 7d). The regression results confirm the relationship between magnetic coercivity and deep-sea redox conditions; specifically, for PSD magnetite, the higher the deep-water oxidation status, the lower the coercivity, and vice versa. In this case, the partial oxidation of the exterior of PSD magnetite grains could reduce the effective diameter and thus reduce the coercivity [27,50].

It is noteworthy that Coe2 only explains ~25% of the variance in Eh values (Figure 7d), and that the coercivity variations are certainly affected by other factors, such as the magnetic grain size, mineralogy, and concentration. It is then suggested magnetic inference for redox conditions cannot be singly made without other evidence.

In addition, the topmost sediments on the seafloor in the western Pacific are typically modern-deposited, while some sites could be affected by random hiatuses [36,51]. Thus, in some cases, the measured samples may contain sediments spanning tens of thousands of years. For those samples, long-term exposure of bottom sediments to oxygen-rich bottom water may have resulted in the prolonged maghemitization of magnetite grains from the shell to the core. In a shell–core modeling study, changes in the coercivity of magnetic grains were positively correlated to the degree of oxidation [27]. Because of this factor, sediments of different ages may represent a source of noise which affects the relationship between magnetic coercivity and deep-water redox conditions.

## 5. Conclusions

We have studied the rock magnetic properties of surficial sediments collected from 66 sites in the western Pacific, with the objective of investigating the relationship between magnetic properties and abyssal redox conditions. The principal results are as follows: (1) Magnetite with PSD magnetic behavior is inferred to be the dominant magnetic mineral in the sediments, and it may have both detrital and biogenic sources. (2) There are no significant differences in the relationships between the measured magnetic parameters. (3) Two major magnetic-coercivity components are identified, with modal coercivity values of $13.1 \pm 1.6$ mT and $54.7 \pm 5.3$ mT, respectively. These components are comparable between hysteresis loops and IRM acquisition curves. Based on these results, we infer that these two magnetic components occur consistently in the bottom sediments across the western Pacific, although their relative proportions vary substantially. All the rock magnetic parameters we measured, including those related to the concentration, grain size, and mineralogy, as well as the magnetic coercivity, are generally correlated to deep-sea sedimentary redox conditions (Eh value), implying a complex connection for magnetic properties. Moreover, coercivities obtained by mathematical unmixing have a significantly negative linkage to redox conditions, accounting for 26% of total variance in redox changes, possibly implying that the partial oxidation could reduce the effective diameter of PSD magnetite grains. Overall, our results demonstrate the potential of magnetic properties of deep-sea sediments in the western Pacific as a convenient and easily measured proxy for changes in sedimentary redox conditions; however, in practice, their use should be validated by other redox indicators.

**Author Contributions:** Conceptualization and methodology, L.Y.; sample collection D.X.; measurements Y.C., Y.L., G.L. and W.C.; formal analysis, L.Y. and H.Q.; original draft preparation, Y.C. All authors contributed to the data interpretation and provided significant input into the final manuscript. All authors have read and agreed to the published version of the manuscript.

**Funding:** This research was funded by National Natural Science Foundation of China, grant number 42177422, and National Programme on Global Change and Air-Sea Interaction, grant number GASI-04-HYDZ-02.

**Institutional Review Board Statement:** Not applicable.

**Informed Consent Statement:** Not applicable.

**Data Availability Statement:** All data supporting the findings of this study are available in an online repository (https://mda.vliz.be/directlink.php?fid=VLIZ_00000824_6339943796b93322139680).

**Acknowledgments:** The authors thank the captain and crew onboard R/V XIANG YANG HONG SHI HAO.

**Conflicts of Interest:** The authors declare no conflict of interest.

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
