# Peer review of "Relationships between Abyssal Redox Conditions and Rock Magnetic Properties of Surficial Sediments in the Western Pacific"

_jmse, doi:10.3390/jmse11061132_

Round 1

Reviewer 1 Report

A review of Chen et al. “A test of the relationship between abyssal redox conditions and rock magnetic properties of surficial sediments in the western Pacific”

Manuscript: jmse-2323724

General comments

Overall, the presented manuscript touches upon an essential topic of redox conditions in deep aquatic settings. Especially important are ways to assess past oxidation and reduction dynamics in the water-sediment interface. Overall, it is a concise manuscript that identifies a research gap and leads to some sensible conclusions. However, the manuscript should have a broader horizon and discussion of the magnetic mineral behavior depending on the redox state.

I’d suggest a shorter ms title: “Relationships between abyssal redox conditions and rock magnetic properties of surficial sediments in the western Pacific.

Once the paper is substantially improved, I see this manuscript as a fine addition to the JMSE.

Specific comments

My main concerns are:

1. Bulk magnetic properties are compared to the Eh state measured immediately after the retrieval, but is this minutiae-scale measurement carried out after the recovery representing Eh conditions on the sea floor?

2. How much of the Eh measurement represents the actual sediment portion, and how much pore water? Is porosity data available to compare potential water influence on the measurement between the samples?

3. Most often, paleoredox proxies are used to estimate (quantitatively and qualitatively) redox conditions on the sediment-water interface or even in the waters overlaying the sediments. Do the authors consider sediment Eh representative of the water Eh?

4. Grain size considerations need to be substantially better explained and introduced in the text.

Introduction

Generally, the introduction needs some reworking, as part of it looks like a random enumeration of available proxies without a clear connection to further reading.

39: a reference?

55-58: random and short paragraph on forams specifically.

74: I think the Authors did not reach this point – when stated like this, I’d expect some water properties data.

Materials and Methods

79-80: needs to be rewritten.

93: 8cm3 from box corer can represent tremendously different sediment age/stratigraphy depending on the considerable depth range of sampling.

Results

Discussion

Generally, the discussion, compared to the introduction, lacks some reflection. I’d expect more discussion on the actual behavior of studied magnetic properties in reference to, e.g., carbon cycling, organic matter decomposition, and so on – things used as a rationale in the introduction.

242-244: statistically speaking, I’m not sure it is an entirely true statement. Correlation between two or more variables and the outcome does not mean a high correlation elsewhere.

246: This is a normative statement, and is the correct figure referenced?

253: grain size is not discussed nor assessed anywhere, so this part looks like a purely literature-based assumption.

264: what constitutes “recent” in this case?

Conclusions

279: this distinction does not seem to come from the Authors study.

292-293: I appreciate this very grounded and accurate conclusion.

Figures and tables

Figure 1: First panel, I suggest using an outline rather than a filled box to show the area. Study sites between the panel are easy to overlook. Maybe put it in the box on the map? There is plenty of space. Depth-color scale is unfortunate and perceptionally not uniform – I recommend, e.g., a cmocean “deep” color palette available for variable software.

Figure 2: technically, these are not biplots (no variable vectors), just simple scatterplots.

Figures 3 and 4: these are not only magnetic properties (Eh). Also, interpolated maps on top of bathymetry are confusing, as potential readers might want to trace the contours between the maps. I suggest showing only the actual interpolation (interest zone) on the maps; therefore, it can also occupy the entire box.

Table 2: Please check journal guidelines, but I’d suggest bolding the significant values rather than putting insignificant values in italics – it is hard to spot at first glance.

No substantial quality issues.

Author Response

A review of Chen et al. “A test of the relationship between abyssal redox conditions and rock magnetic properties of surficial sediments in the western Pacific”

Manuscript: jmse-2323724

General comments

Overall, the presented manuscript touches upon an essential topic of redox conditions in deep aquatic settings. Especially important are ways to assess past oxidation and reduction dynamics in the water-sediment interface. Overall, it is a concise manuscript that identifies a research gap and leads to some sensible conclusions. However, the manuscript should have a broader horizon and discussion of the magnetic mineral behavior depending on the redox state.

I’d suggest a shorter ms title: “Relationships between abyssal redox conditions and rock magnetic properties of surficial sediments in the western Pacific.

Reply: agree and revised accordingly.

Once the paper is substantially improved, I see this manuscript as a fine addition to the JMSE.

Reply: agree and revised accordingly.

  • We changed figures accordingly
  • We revised the text to make it clearer
  • We mentioned the potential flaws in the revision

We thank you for your professional reviewing our paper, and the revision has been significantly improved from your comments.

Specific comments

My main concerns are:

  1. Bulk magnetic properties are compared to the Eh state measured immediately after the retrieval, but is this minutiae-scale measurement carried out after the recovery representing Eh conditions on the sea floor?

Reply: agree and revised accordingly.

We have mentioned in the section 2.1

Lines 79-80:

The redox potential (Eh value) of the sediments was immediately measured on board, us-ing a portable pH meter (INESA Co., PHBJ-260).

  1. How much of the Eh measurement represents the actual sediment portion, and how much pore water? Is porosity data available to compare potential water influence on the measurement between the samples?

Reply: agree and revised accordingly.

We mentioned that in the revision that Eh values are sedimentary redox conditions, as your suggestion. And in the main text, they all have changed to “sedimentary redox conditions”

Lines 81-83:

Since the measurements were conducted in an on-board lab, the obtained Eh values may not directly represent the real ones of deep-sea environment, but a proxy of sedimentary redox conditions.

  1. Most often, paleoredox proxies are used to estimate (quantitatively and qualitatively) redox conditions on the sediment-water interface or even in the waters overlaying the sediments. Do the authors consider sediment Eh representative of the water Eh?

Reply: agree and revised accordingly.

We mentioned that in the revision that Eh values are sedimentary redox conditions, as your suggestion. And in the main text, they all have changed to “sedimentary redox conditions”

Lines 81-83:

Since the measurements were conducted in an on-board lab, the obtained Eh values may not directly represent the real ones of deep-sea environment, but a proxy of sedimentary redox conditions.

  1. Grain size considerations need to be substantially better explained and introduced in the text.

Reply: Grain-size of magnetic minerals is shown in Fig. 5c, while for the precise observation, it is dependent on detailed TEM analysis, which is not available for our study. Thus, we did not expand this part in the revision.

Introduction

Generally, the introduction needs some reworking, as part of it looks like a random enumeration of available proxies without a clear connection to further reading.

39: a reference?

Reply: agree and revised accordingly.

55-58: random and short paragraph on forams specifically.

Reply: agree and deleted accordingly.

74: I think the Authors did not reach this point – when stated like this, I’d expect some water properties data.

Reply: agree and deleted accordingly.

Materials and Methods

79-80: needs to be rewritten.

Reply: agree and revised accordingly.

Lines 77-78

Sediment samples in 66 sites were collected in the western Pacific (Fig. 1) in October 2017, using a box corer.

93: 8cm3 from box corer can represent tremendously different sediment age/stratigraphy depending on the considerable depth range of sampling.

Reply: the 8 cm3 is a typical size for environmental magnetic measurements. We mentioned in the revision that it may represent tens of thousands of years.

Line 275

The measured samples may contain sediments spanning tens of thousands of years.

Results

Discussion

Generally, the discussion, compared to the introduction, lacks some reflection. I’d expect more discussion on the actual behavior of studied magnetic properties in reference to, e.g., carbon cycling, organic matter decomposition, and so on – things used as a rationale in the introduction.

Reply: we agree that these issues you mentioned were not mentioned in the draft. The reason is because the paper is dealing with surface sediments not borehole sediments. For surface sediments, they were collected during a geological cruise, and thus, no other indices of water properties were obtained. Discussion about environmental changes is usually dependent on borehole sediments. These environmental issues will be considered when we get gravity cores in future. Thanks for your advice.

242-244: statistically speaking, I’m not sure it is an entirely true statement. Correlation between two or more variables and the outcome does not mean a high correlation elsewhere.

Reply: agree and deleted accordingly.

246: This is a normative statement, and is the correct figure referenced?

Reply: agree and revised accordingly.

253: grain size is not discussed nor assessed anywhere, so this part looks like a purely literature-based assumption.

Reply: agree and revised accordingly.

Lines 259-262

In addition, different grain sizes of magnetic minerals may have different behaviors responding to oxidation [27], which could explain a weak correlation between Eh and Bcr/Bc (Fig. 7c), since the Bcr/Bc ratio can be a proxy of magnetic grain size [49], like χfd, χARM/χ, SIRM/χ, and χARM/SIRM.

264: what constitutes “recent” in this case?

Reply: agree and revised accordingly.

Lines 273-274

In addition, although the topmost sediments on seafloor in the western Pacific are typically modern-deposited, hiatuses may exist randomly at some sites [36,51].

Conclusions

279: this distinction does not seem to come from the Authors study.

Reply: agree and revised accordingly.

  • We add x-T curves in the revision to support that the major magnetic mineral is magnetite
  • We have TEM to indicate the possible presence of magnofossils

Lines 231-233

The TEM investigation of Site J01A (Fig. 5f) shows chain-like shape of magnetic minerals, confirming that magnetofossils are widespread in the abyssal sediments of the western Pacific.

Lines 287-288

Magnetite with PSD magnetic behavior is inferred to be the dominant magnetic mineral in the sediments, and it may have both detrital and biogenic sources.

292-293: I appreciate this very grounded and accurate conclusion.

Thanks again for your professional comments.

Figures and tables

Figure 1: First panel, I suggest using an outline rather than a filled box to show the area. Study sites between the panel are easy to overlook. Maybe put it in the box on the map? There is plenty of space. Depth-color scale is unfortunate and perceptionally not uniform – I recommend, e.g., a cmocean “deep” color palette available for variable software.

Reply: agree and revised accordingly.

Figure 2: technically, these are not biplots (no variable vectors), just simple scatterplots.

Reply: agree and revised accordingly.

Figures 3 and 4: these are not only magnetic properties (Eh). Also, interpolated maps on top of bathymetry are confusing, as potential readers might want to trace the contours between the maps. I suggest showing only the actual interpolation (interest zone) on the maps; therefore, it can also occupy the entire box.

Reply: agree and revised accordingly.

Table 2: Please check journal guidelines, but I’d suggest bolding the significant values rather than putting insignificant values in italics – it is hard to spot at first glance.

Reply: agree and revised accordingly.

The revised Table 3 (table 2 in the submission) is now following the journal format.

Reviewer 2 Report

I believe that the text of the manuscript needs improvement.

There are two fundamental points.

1. The statement about the predominance of magnetite in precipitation needs additional evidence. These can be either thermomagnetic or microprobe analyses.,

2. It is necessary to make either a more cautious or a more evidence-based conclusion about the connection of the magnetic parameters established by the authors with the redox conditions in the waters and at the bottom of the studied part of the Pacific Ocean.

«All the rock magnetic parameters we measured, including those related to the concentration, grain size and mineralogy, as well as the magnetic coercivity, are generally correlated to deep-water redox conditions (Eh value), and that coercivities obtained by mathematical unmixing have a stronger linkage to redox conditions than the other magnetic parameters».

These and other comments and suggestions are given in the margins of the manuscript.

Author Response

I believe that the text of the manuscript needs improvement.

We thank you for your professional reviewing our paper, and the revision has been significantly improved from your comments.

There are two fundamental points.

  1. The statement about the predominance of magnetite in precipitation needs additional evidence. These can be either thermomagnetic or microprobe analyses.,

Reply: agree and revised accordingly.

  • TEM analysis is presented in Fig. 5f to show a chain-like structure
  • X-T curves are added in Fig. 5g-i

  1. It is necessary to make either a more cautious or a more evidence-based conclusion about the connection of the magnetic parameters established by the authors with the redox conditions in the waters and at the bottom of the studied part of the Pacific Ocean.

«All the rock magnetic parameters we measured, including those related to the concentration, grain size and mineralogy, as well as the magnetic coercivity, are generally correlated to deep-water redox conditions (Eh value), and that coercivities obtained by mathematical unmixing have a stronger linkage to redox conditions than the other magnetic parameters».

These and other comments and suggestions are given in the margins of the manuscript.

Reply: agree and revised accordingly.

Lines 294-301

All the rock magnetic parameters we measured, including those related to the concentration, grain size and mineralogy, as well as the magnetic coercivity, are generally correlated to deep-sea sedimentary redox conditions (Eh value), inferring a complex connection for magnetic properties. Moreover, coercivities obtained by mathematical unmixing have a significantly negative linkage to redox conditions, accounting for ~25% of total variance in redox changes, possibly inferring that the partial oxidation could reduce the effective diameter of PSD magnetite grains.

Comments in the PDF file

  1. Line 144 It is necessary to specify the number of samples measured. Section 2.1 only states that samples were taken from 66 sites. And what is the number of samples?

Reply: agree and revised accordingly.

  1. Line 152 surface à bottom

Reply: agree and revised accordingly.

  1. Lines 154-155 In Fig. 2 it can be seen that the studied set contains several dozen samples. I believe that: 1) it is necessary to specify the exact number of samples examined here; 2) in the note to Table. 2 give the exact value of the significance level at p = 0.05.

Reply: agree and revised accordingly.

  • The number of the examined samples are listed.
  • The table has revised to follow the journal format.

  1. Line 176 There are not only magnetic parameters here.

Reply: agree and revised accordingly.

  1. Line 187 A similar remark is made as to the caption to Fig. 3. There are not only parameters of magnetic coercivity here. And a general note to Fig. 3 and Fig. 4. Too small drawings with a meaningful part (rectangles with drawing of isolines).

Reply: agree and revised accordingly.

  1. Lines 195-196 This is a clear stretch. I think this phrase is superfluous.

Reply: agree and deleted accordingly.

  1. Line 204 surface à bottom

Reply: agree and revised accordingly.

  1. Line 211 It is necessary to specify the number of measured samples and model calculations.

Reply: agree and revised accordingly.

  1. Lines 234-235 I believe that for absolute certainty in this statement, it is necessary to make several thermomagnetic analyses - saturation magnetization from temperature(Ms-T).

Reply: agree and revised accordingly. X-T curves were added in the revision.

  1. Line 236 surface à bottom

Reply: agree and revised accordingly.

  1. Lines 240-244 First, in Fig. 7d the correlation coefficient is not specified, but the coefficient of determination. Secondly, and this is the main thing, the Core2-Eh determination coefficient is 0.26, and according to the Robert Emmet Chaddock classification, this relationship is estimated as weak. I think it is necessary to give explanations in the text. Number of samples?

Reply: 1) The explained variance (R2) is listed in Fig. 7d, and in other subplots, it is correlation coefficient. Thus, the correlation coefficient in Fig. 7d is 0.52, not weak.

2) We enlarge the labels in Fig. 7d to make more readable

3) The sample size is listed in the revision.

  1. Lines 255-257 I emphasize once again, according to Fig. 7d, the correlation here is weak.

Reply: The correlation coefficient in Fig. 7d is 0.52, and we enlarge the labels to make more readable

  1. Lines 278-280 For a more evidence-based statement, either thermomagnetic analysis or microprobe analysis is needed.

Reply: agree and revised accordingly.

  • We add x-T curves in the revision to support that the major magnetic mineral is magnetite
  • We have TEM to indicate the possible presence of magnofossils

Lines 231-233

The TEM investigation of Site J01A (Fig. 5f) shows chain-like shape of magnetic minerals, confirming that magnetofossils are widespread in the abyssal sediments of the western Pacific.

Lines 287-288

Magnetite with PSD magnetic behavior is inferred to be the dominant magnetic mineral in the sediments, and it may have both detrital and biogenic sources.

  1. Lines 286-290 I think such an unambiguous conclusion is premature.

Reply: agree and revised accordingly.

Lines 296-301

All the rock magnetic parameters we measured, including those related to the concentration, grain size and mineralogy, as well as the magnetic coercivity, are generally correlated to deep-sea sedimentary redox conditions (Eh value), inferring a complex connection for magnetic properties. Moreover, coercivities obtained by mathematical unmixing have a significantly negative linkage to redox conditions, accounting for ~25% of total variance in redox changes, possibly inferring that the partial oxidation could reduce the effective diameter of PSD magnetite grains.

Round 2

Reviewer 1 Report

A review of Chen et al. “Relationships between abyssal redox conditions and rock magnetic properties of surficial sediments in the western Pacific”

Manuscript: jmse-2323724

General comments

Generally, the authors addressed my issues, and the manuscript has improved. I strongly suggest that before acceptance, the manuscript goes through language editing, especially parts of the text that were recently added. A few minor comments are below, but once the Authors do the final corrections, I don’t see the need for another round of review.

Specific comments

38-40: this sentence needs some grammar corrections.

Fig 1 caption: Schematic map showing the study area and study sites. – remove the number.

Tab. 1: Site – variable in singular; add unit to latitude and longitude (decimal degrees?).

Figure 2 caption and later figures (3, 4, 5, and so on) should probably sound more like this: It is noted that there were in total 66 samples analyzed. But overall, I’m unsure if writing it down for every Figure and Table caption is essential.

Carefully check for spaces between the numbers and units (it is not consistent throughout the text)

254: 0.05 is a rather good threshold.

274: typically, modern-deposited, but some sites could be affected by the random hiatuses...

Minor but thorough editing is necessary. I suggest that the Authors use publisher's or outside company services for that.

Author Response

A review of Chen et al. “Relationships between abyssal redox conditions and rock magnetic properties of surficial sediments in the western Pacific”

Manuscript: jmse-2323724

General comments

Generally, the authors addressed my issues, and the manuscript has improved. I strongly suggest that before acceptance, the manuscript goes through language editing, especially parts of the text that were recently added. A few minor comments are below, but once the Authors do the final corrections, I don’t see the need for another round of review.

Reply: agree and revised accordingly. The revision was checked and typos were removed.

Specific comments

38-40: this sentence needs some grammar corrections.

Reply: agree and the sentence was rewritten.

Lines 39-40

the proxies are mainly based on geochemical properties of foraminiferal shells or deep-sea sediments

Fig 1 caption: Schematic map showing the study area and study sites. – remove the number.

Reply: agree and revised accordingly.

Tab. 1: Site – variable in singular; add unit to latitude and longitude (decimal degrees?).

Reply: agree and revised accordingly.

Figure 2 caption and later figures (3, 4, 5, and so on) should probably sound more like this: It is noted that there were in total 66 samples analyzed. But overall, I’m unsure if writing it down for every Figure and Table caption is essential.

Reply: agree and revised accordingly.

Carefully check for spaces between the numbers and units (it is not consistent throughout the text)

Reply: agree and checked.

254: 0.05 is a rather good threshold.

Reply: thanks.

274: typically, modern-deposited, but some sites could be affected by the random hiatuses

Reply: agree and revised accordingly.
